# Positive Status Disclosure and Sexual Risk Behavior Changes among People Living with HIV in the Northern Region of Ghana

Peter Claver Kabriku [1], Edward Wilson Ansah [2] and John Elvis Hagan, Jr. [2,3,*]

1    Tamale Nursing and Midwifery Training College, Tamale, Ghana; pkabriku@gmail.com
2    Department of Health, Physical Education and Recreation, University of Cape Coast, Cape Coast, Ghana; edward.ansah@ucc.edu.gh
3    Neurocognition and Action-Biomechanics-Research Group, Faculty of Psychology and Sport Sciences, Bielefeld University, Postfach 10 01 31, 33501 Bielefeld, Germany
*    Correspondence: elvis.hagan@uni-bielefeld.de

**Abstract:** Objective: To investigate sexual behavior changes adopted by People Living with Human Immunodeficiency Virus (PLHIV) on Antiretroviral therapy (ART) in the Northern Region of Ghana. Methods: We employed a cross-sectional survey with a questionnaire to collect data from 900 clients from 9 major ART centers within the region. Chi-square and logistic regression analyses were applied to the data. Results: More than 50% of PLHIV on ART use condoms, reduce sexual partners, practice abstinence, reduce unprotected sex with married/regular partners, and avoid casual sex. Fear of others getting to know patients' HIV-positive status ($\chi^2$ = 7.916, $p$ = 0.005), stigma ($\chi^2$ = 5.201, $p$ = 0.023), and fear of loss of family support ($\chi^2$ = 4.211, $p$ = 0.040) significantly predict non-disclosure of HIV-positive status among the participants. Change in sexual behavior is influenced by the following: "to avoid spreading the disease to others" ($R^2$ = 0.043, $F$ (1, 898) = 40.237, $p$ < 0.0005), "to avoid contracting other STIs" ($R^2$ = 0.010, $F$ (1, 898) = 8.937, $p$ < 0.0005), "to live long" ($R^2$ = 0.038, $F$ (1, 898) = 35.816, $p$ < 0.0005), "to hide HIV-positive status" ($R^2$ = 0.038, F (1, 898) = 35.587, $p$ < 0.0005), "to achieve good results from ART treatment" ($R^2$ = 0.005, $F$ (1, 898) = 4. 282, $p$ < 0.05), and "to live a Godly life" ($R^2$ = 0.023, $F$ (1, 898) = 20. 880, $p$ < 0.0005). Conclusions: High self-disclosure rate of HIV-positive status was identified, with participants disclosing to their spouses or parents. Reasons for disclosure and non-disclosure differed from person to person.

**Keywords:** PLHIV; disclosure status; sexual behavior change; ART; Ghana

## 1. Introduction

Human Immunodeficiency Virus (HIV) and the Acquired Immune Deficiency Syndrome (AIDS) pandemic still pose global public health challenges despite decades of numerous interventions aimed at eradicating the disease [1]. HIV is the virus that causes AIDS, the terminal stage of HIV infection. The virus can be acquired through heterosexual contact, mother-to-child (MTC) transmission, and contaminated blood transfusion [2,3]. Apart from vertical transmission, unprotected sex is the most common route of HIV infection among people living with HIV (PLHIV), while the route of infected needles sharing is the least [3]. However, a recent study showed that more males contracted HIV through narcotics injection, while the females contracted it through unprotected sexual contact with infected individuals [4].

The world continues to implement HIV-prevention strategies to improve the quality of life of HIV-infected persons and their sexual and social networks [5]. HIV disclosure among HIV-infected persons is one of these strategies considered effective for early management initiatives [6]. Disclosure of HIV-positive status is an adopted concept that has its roots in self-disclosure, a communication style in which an individual shares information about

their lives (i.e., positive HIV status) [7]. There is evidence that disclosing one's HIV status affects one's behavior. For example, a study on the HIV Prevention of Mother to Child Transmission (PMTC) Program revealed that among HIV-positive mothers, disclosing HIV-positive status to family members, especially sexual partners, improved adherence to exclusive newborn feeding [8].

The AIDS risk reduction model (ARRM) provides a framework for explaining and predicting behavioral change efforts of persons with STIs such as HIV/AIDS [9]. This model operates around three tenants: recognition and labeling, commitment, and action-taking. In addition, the health belief model (HBM) explains the concepts that predict the reasons individuals take action to prevent, screen for, or control illness conditions [10]. It operates with primary concepts of perception: susceptibility, severity, benefits, and barriers to behavior; cues to action; and self-efficacy. Thus, in HIV prevention and management, individuals need to recognize and acknowledge the behaviors and label such actions as risky and take action accordingly. Besides, HIV-infected individuals must feel susceptible to infecting others and reinfecting themselves; understand the severity of the disease on their physical, social, and economic life; understand the benefits and barriers; and believe in their ability to make the needed behavioral changes in managing the HIV condition. Within the context of HIV/AIDS prevention and management, HIV disclosure is paramount. For instance, HIV status disclosure has a significant role in new HIV infection prevention and management [11]. Some studies have revealed that HIV-positive status disclosure can bring about emotional/financial support and increase partner HIV testing and acceptance of condom use [12,13]. Moreover, HIV disclosure is important in the context of ART, which can lead to care support and treatment adherence [14].

Othher empirical evidence further shows that many people engage in risky sexual behavior, which puts them at risk of getting themselves and others infected with HIV and other STIs [15]. Previous studies have revealed that over 50% of PLHIV are sexually active and engage in sexual risk behavior even when they are on ART [16,17]. Comparatively, PLHIV and those on ART change their sexual risk behavior [18,19]. For instance, it has been established that PLHIV on ART engage in consistent condom use, reduce the number of sexual partners, or help in partner(s) HIV testing due to the counseling they receive before and after initiating the ART [16]. Again, Bolori, Mohammed Tahiru et al. [20] found that approximately 83% of respondents had an attitudinal change after being diagnosed as HIV-positive. Such changes include taking preventive measures against more serious STIs, avoiding the spread of the disease to others, and complying with medical advice on a regular basis.

HIV/AIDS prevalence in Ghana still poses a public health concern despite national efforts to curb local transmission. Therefore, in 2013, Ghana's Ministry of Health, in collaboration with the Ghana AIDS Commission, implemented enhanced early access to ART at no cost to PLHIV regardless of the patient's immunologic status or viral load [21–23]. Thus, the successful implementation of strategies to reach those at risk is crucial for decreasing infection in the country. However, the prevalence of HIV keeps rising; it rose from 1.67% in 2017 to 1.69 in 2018, with regional variations. During the same period, the Northern Region saw a change of 100% (0.2% to 0.4%) [24]. The national estimate was 1.7%, with about 342,307 people living with the disease in 2019. Moreover, there were about 20,068 new HIV infections and an estimated 13,616 HIV-related deaths for the same period [22]. Out of the new infections, about 5532 (28%) were between the ages of 15 and 24. The Bono Region recorded the highest prevalence of 2.66%, representing 20,134 PLHIV in the region, while the North East Region recorded the lowest prevalence of 0.39%. Unfortunately, only about 61% of PLHIV were on ART as of 2018 [22], with 2896 in the Northern Region in 2019 [24]. This trend suggests that people in the area are infected more frequently with HIV, which needs more attention. However, most HIV studies have investigated serological status disclosure within-between specific sub-groups, such as men who have sex with men (MSM) [25], bisexual men [26], or women [27,28], but have not explicitly assessed disclosure

and behavioral patterns among subgroups of PLHIV and across different populations with varied socio-cultural contexts.

Therefore, the purpose of this study was to explore the sexual behavior changes adopted by PLHIV on ART in the Northern Region and identify factors motivating such changes in sexual behavior among the patients.

## 2. Materials and Methods

### 2.1. Participants' Selection Criteria

There are twenty-nine ART centers in the Northern Region of Ghana that offer both therapy and counseling services to PLHIV as of 2020. Out of these, nine were considered major centers according to the regional HIV data manager. The nine major centers include the Tamale Teaching Hospital, Tamale Central Hospital, Tamale West Hospital, Bimbilla, Saboba, Zabzugu, Yendi, Gushiegu, and Savelugu Hospitals, which were selected using a purposive sampling method for the study. These nine facilities were chosen because they are district hospitals or higher and provide ART services to patients in the region. Again, these centers have a lot of registered clients taking their treatment at the facilities. Moreover, most of the clients at these facilities had longer periods of therapy, and they may have had rich experiences to share during the study [24].

There were approximately 2896 adults with PLHIV who registered for ART in these nine centers as of October 2019 [24]. Therefore, an accidental sampling was used to recruit 900 patients, a response rate of 31.1%, between August and September 2020, a figure that represents a little above 31% of the population. The selection of the participants took place at the ART centers. With the help and consent of the facility heads and the patients, we administered the questionnaire to the available patients at the various centers. This sample size was affected by the COVID-19 restrictions in the year 2020, which led to a reduced number of participants visiting the ART centers. However, this sample size is adequate for a survey for generalization to the PLHIV on ART in the Northern Region. Moreover, this is also a very sensitive study on a phenomenon such as HIV and AIDS patients [28,29].

Many of the patients possess no formal education; thus, the instrument was administered with the help of two trained research assistants who assisted the research participants in filling out the questionnaire. These research assistants completed their bachelor's degree in nursing and were trained to translate the instrument into Dabgani (a local language spoken by the people of Tamale) and back-translated it into English language. This translation method was used to minimize variations in the translation of the instrument and limit the introduction of bias from the research assistants.

### 2.2. Instrumentation

A 32-item questionnaire was developed based on the literature and put into 3 sections (A, B, and C). Section A had 10 items that collected data on socio-demographic characteristics of the patients, such as age, gender, level of education, religious affiliation, marital status, duration of sexual relationship, employment status, sexual partner's HIV status, place of residence, and participants' current number of sexual partners. Participants responded to these items using a multiple-choice scale. Section B contained 13 items that solicited information on respondents' sexual risk behavior change and the factors influencing such behavioral change after they were diagnosed HIV-positive. This section was measured on a 4-point Likert scale of always (4) to never (1), and strongly agree (4) to strongly disagree (1). Section C comprised 9 items that obtained information on HIV diagnosis, treatment, prevention, and positive status disclosure/non-disclosure process. Participants responded to these items using a dichotomous scale (Yes or No) as well as multiple-choice options. The instrument is deposited at Open Science Framework and can be found here https://osf.io/b49ta/ (accessed on 30 April 2023).

The questionnaire was evaluated by three patients (who did not take part in the main study) and two senior research assistants. The instrument was later assessed by two senior faculty members in Health Promotion from the University of Cape Coast. Pretesting the

instrument was completed at the Cape Coast Teaching Hospital using 35 PLHIV who were on ART because the hospital also handles clients similar to those in the Northern Region. The questionnaire yielded a reliability coefficient (Cronbach alpha) of 0.76. Specifically, respondents' sexual risk behavior change and the influencing factors yielded 0.72, and information on HIV diagnosis, treatment, prevention, and positive status disclosure/non-disclosure process gave 0.75.

Ethical approval was obtained from the Institutional Review Board (IRB) at the University of Cape Coast, Ghana (UCCIRB/CES/2020/23). We also sought and gained permission from the Northern Regional Director of Health Services and the heads of all the ART centers. Moreover, each participant either gave written or verbal consent before taking part in the study.

*2.3. Data Analysis*

Descriptive statistics (i.e., frequencies and percentages) were used to analyze the proportion of HIV-positive status disclosure among the patients. Additionally, Chi-square was calculated to assess gender differences in HIV status disclosure percentage and sexual behavior change adopted by PLHIV on ART in the Northern Region. A simple regression was used to determine factors influencing non-disclosure of HIV-positive status by PLHIV on ART. Additionally, binary logistic regression analysis determined the factors influencing sexual behavior change among the patients after their positive status diagnosis. The statistical significance was set at a 95% confidence interval and *p*-values less than 0.05 ($p < 0.05$).

**3. Results**

The results are presented according to the demographic characteristics of participants, the positive status disclosure rate and gender difference in disclosure, reasons for non-disclosure, adopted sexual risk behavior changes, and the factors influencing change in risky sexual behavior among the patients.

*3.1. Demographic Characteristics of Participants*

Out of 900 HIV patients on ART in this study, 658 (73.1%) were females, and 242 (26.9%) were males. Overall, 387 (43.0%) of the participants had no formal education, and a few, 72 (8%), were single and had never married. Moreover, most of the patients (n = 396; 44%) were unemployed, and 54 (6%) were students. In addition, the majority of the participants (n = 490; 54.4%) who were married or in sexual relationships did not know their partners' HIV status. Meanwhile, 284 (31.6%) had been married or in a sexual relationship for 10 years or more. Slightly more than half (n = 452; 50.2%) of the participants were urban dwellers, and most 689 (76.6%) indicated having only one sexual partner.

*3.2. Percentage of HIV-Positive Status Disclosure among PLHIV on ART in the Northern Region*

Different percentages of HIV patients made positive status disclosure to different people. For instance, out of the 900 patients on ART, 540 (60%) disclosed their HIV-positive status to other people. Specifically, of those who disclosed, 52% disclosed to their spouse (wife or husband), while a few (39%) disclosed to their siblings and approximately 37% to their parents. However, a very small number (less than 14%) of the patients disclosed to their co-workers; employers; religious leaders; and other relatives, such as aunts, uncles, and grandchildren (see Table 1).

*3.3. Gender Difference in HIV-Positive Status Disclosure*

The results further showed no statistically significant difference between males and females in their HIV-positive self-disclosure status: $\chi^2$ (1, n = 900) = 0.30, *p* = 0.58, phi = 0.02. Even though the percentages of positive status disclosure were slightly greater for males (61.6%) than female respondents (59.3%), the Chi-square analysis indicated that the gender difference in terms of the positive status disclosure proportion was not significant

statistically (see Table 2). This implies that during counseling and treatment periods, both males and females on ART in the Northern Region are almost equally likely to tell someone about their HIV-positive status. The results also revealed that 40% of the PLHIV did not self-disclose their positive status to any other persons.

**Table 1.** Percentage of HIV-positive status disclosure or non-disclosure among PLHIV on ART in Northern Region.

| Variable | Yes *f* (%) | No *f* (%) |
|---|---|---|
| Self-disclosure of HIV-positive status (n = 900) | 540 (60) | 360 (40) |
| Disclosure to parents (both or either) (n = 540) | 198 (36.7) | 342 (63.3) |
| Disclosure to husband/wife (n = 540) | 280 (51.9) | 260 (48.1) |
| Disclosure to siblings (n = 540) | 209 (38.7) | 331 (61.3) |
| Disclosure to present sexual partner(s) (n = 538) | 8 (1.5) | 530 (98.5) |
| Disclosure to co-workers (n = 538) | 4 (0.7) | 534 (99.3) |
| Disclosure to employer (n = 538) | 2 (0.4) | 536 (99.6) |
| Disclosure to religious leader (n = 538) | 72 (13.4) | 466 (88.6) |

Keys: n = number of respondents; HIV = Human Immunodeficiency Virus.

**Table 2.** Gender-stratified percentage distribution of HIV-positive status disclosure.

| Disclosure of HIV Status | Males (n = 242) | | Female (n = 658) | | $\chi^2$ (df) | *p*-Value |
|---|---|---|---|---|---|---|
| | n | % | n | % | | |
| Yes | 93 | 61.6 | 390 | 59.3 | 0.30 (1) 0.58 | 0.02 * |
| No | 149 | 38.4 | 268 | | | |

Keys: n = number of respondents; HIV = Human Immunodeficiency Virus; * $p < 0.05$, yes = number of patients who disclosed their HIV posotive status, no = those who did not disclose their HIV positive status

### 3.4. Reasons for Non-Disclosure of HIV-Positive Status

Results from logistics regression revealed that the model was statistically significant in determining the non-disclosure status of the patients, overall model (constant) ($\chi^2$ (11, N = 360) = 1063.59, $p < 0.001$). Specifically, fear of others getting to know patients' HIV-positive status ($\chi^2$ = 7.916, $p = 0.005$), fear of stigma ($\chi^2$ = 5.201, $p = 0.023$), fear of losing family support ($\chi^2$ = 4.211, $p = 0.040$) and "not ready to disclose yet" ($\chi^2$ = 4.475, $p = 0.034$) were statistically significant predictors of non-disclosure of HIV-positive status among the participants. However, fear of verbal/physical abuse ($p = 0.289$), fear of losing a marriage/relationship ($p = 0.224$), and fear of losing a job ($p = 0.554$) were not statistically significant in predicting HIV status non-disclosure among the participants (See Table 3).

### 3.5. Sexual Behavior Changes Adopted by Participants after Learning Their HIV Status

The results again showed that 492 (54.6%) of the patients used condoms always or sometimes during sexual intercourse, while 408 (45%) never or rarely used condoms during sexual intercourse. Furthermore, 344 (38%) never or rarely reduced the number of sexual partners they had, whereas 556 (62.8%) did so as they learned of their HIV-positive status. Moreover, more than half, 551 (61%), of the participants always or sometimes abstained from sexual intercourse, while 349 (39%) never or rarely abstained. Again, the majority, 586 (65%), of participants indicated that they never or rarely engaged in unprotected sexual intercourse with their regular/married partners, while 314 (35%) always or sometimes engaged in unprotected sexual intercourse with such partners. Moreover, a little over half, 465 (52%), of the participants reported they never or rarely indulged in casual sexual intercourse, while 435 (48%) did always or sometimes (see Table 4 for details). As a result, more than half of PLHIV on ART in the Northern Region have adopted safe or positive

sexual behaviors, such as the use of condoms, a reduction in the number of sexual partners, abstinence, a reduction in unprotected sex with married/regular partners, and avoidance of new casual sex after learning of their HIV-positive status.

**Table 3.** Predictors of non-disclosure of HIV-positive status by PLHIV on ART in Northern Region (N = 360).

| Variables | Wald | Df | Sig | Odds Ratio | 95% C. I. |
|---|---|---|---|---|---|
| Fear of rejection by family | 0.337 | 1 | 0.562 | 0.646 | 0.147–2.829 |
| Fear of verbal/physical abuse | 1.125 | 1 | 0.289 | 3.478 | 0.348–34.799 |
| Fear of losing a marriage/relationship | 1.478 | 1 | 0.224 | 0.377 | 0.078–1.817 |
| Fear of many others getting to know my HIV status | 7.916 | 1 | **0.005** | 0.019 | 0.001–0.301 |
| Fear of stigma from others | 5.201 | 1 | **0.023** | 0.079 | 0.009–0.699 |
| Fear of losing family support | 4.211 | 1 | **0.040** | 20.702 | 1.146–374.040 |
| Fear of losing job | 0.350 | 1 | 0.554 | 0.556 | 0.079–3.891 |
| Not ready to disclose yet | 4.475 | 1 | **0.034** | 0.176 | 0.035–0.880 |
| Constant | 1.916 | 1 | 0.166 | 3.136 | 0.001 |

Bold: Sig. < 0.05.

**Table 4.** Frequency data on sexual behavior changes adopted by participants three months after positive status awareness.

| Variables | Never f (%) | Rarely f (%) | Sometimes f (%) | Always f (%) |
|---|---|---|---|---|
| Condom use during sexual intercourse | 273 (30.3) | 135 (15) | 183 (20.3) | 309 (34.3) |
| Reduction in number of sexual partners | 213 (23.7) | 131 (14.6) | 203 (22.6) | 353 (39.2) |
| Abstinence from sexual intercourse | 215 (23.9) | 134 (14.9) | 298 (33.1) | 253 (28.1) |
| Unprotected sex only with married/regular partner(s) | 438 (48.7) | 148 (16.4) | 144 (16) | 170 (18.9) |
| Avoidance of new casual sexual intercourse | 402 (44.7) | 63 (7) | 105 (11.7) | 330 (36.7) |

### 3.6. Factors Influencing Changes in Risky Sexual Behavior of PLHIV and on ART

The results of regression analysis showed that the model containing all the predictors was statistically significant and yielded 3.9% of the variance in the sexual behavior change of patients who knew of their HIV-positive status ($R^2 = 0.039$, $F (1, 898) = 36.268$, $p < 0.05$). However, the most significant independent factor that influenced a change in sexual behavior (DV) was "to avoid spreading the disease to others", ($R^2 = 0.043$, $F (1, 898) = 40.237$, $p < 0.0005$), explaining 4.3% of the variance in sexual behavior change among the patients.

The result again showed that the lowest (.4%) independent predictor of change in sexual risk behavior was "to gain support from family and others" ($R^2 = 0.004$, $F (1, 898) = 3.988$, $p < 0.05$). The rest of the factors that significantly influenced a change in sexual behavior were: "to avoid contracting other STIs" ($R^2 = 0.010$, $F (1, 898) = 8.937$, $p < 0.0005$), "live long" ($R^2 = 0.038$, $F (1, 898) = 35.816$, $p < 0.0005$), "hide HIV-positive status" ($R^2 = 0.038$, $F (1, 898) = 35.587$, $p < 0.0005$), "achieve good results from ART treatment" ($R^2 = 0.005$, $F (1, 898) = 4.282$, $p < 0.05$), and "live a Godly life" ($R^2 = 0.023$, $F (1, 898) = 20.880$, $p < 0.0005$). However, "to avoid impregnating women or getting pregnant" was not a statistically significant predictor ($p = 0.173$) of the change in sexual behavior among patients on ART (see Table 5).

**Table 5.** Contribution of individual factors to the prediction of change in risky sexual behaviors of PLHIV (N = 900).

| Model Predictors | Unstandardized Coefficients | | Standardized Coefficients | | |
|---|---|---|---|---|---|
| | B | Std. Error | Beta | t | *p*-Value |
| 1 (constant) | 1.862 | 0.102 | 18.304 | | 0.000 |
| To avoid spreading the disease to others | 0.192 | 0.030 | 0.207 | 6.343 | 0.000 |
| Model summary: $R^2$ = 0.043, F (1, 898) = 40.237, *p* < 0.0005 | | | | | |
| 1 (constant) | 2.179 | 0.107 | 20.288 | | 0.000 |
| To avoid contracting other STIs | 0.097 | 0.032 | 0.099 | 2.990 | 0.000 |
| Model summary: $R^2$ = 0.010, F (1, 898) = 8.937, *p* < 0.0005 | | | | | |
| 1 (constant) | 2.117 | 0.066 | 31.931 | | 0.000 |
| To live long | 0.128 | 0.021 | 0.196 | 5.985 | 0.000 |
| Model summary: $R^2$ = 0.038, F (1, 898) = 35.816, *p* < 0.0005 | | | | | |
| 1 (constant) | 2.155 | 0.060 | 350.669 | | 0.000 |
| To hide my HIV-positive status | 0.123 | 0.021 | 0.195 | 5.966 | 0.000 |
| Model summary: $R^2$ = 0.038, F (1, 898) = 35.587, *p* < 0.0005 | | | | | |
| 1 (constant) | 2.300 | 0.096 | 23.946 | | 0.000 |
| To achieve good results from ART | 0.060 | 0.029 | 0.069 | 2.069 | 0.000 |
| Model summary: $R^2$ = 0.005, F (1, 898) = 4.282, *p* < 0.0005 | | | | | |
| 1 (constant) | 2.427 | 0.058 | 38.842 | | 0.000 |
| To live a Godly life | 0.094 | 0.020 | 0.151 | 4.569 | 0.039 |
| Model summary: $R^2$ = 0.023, F (1, 898) = 20.880, *p* < 0.0005 | | | | | |
| 1 (constant) | 2.606 | 0.060 | 43.491 | | 0.000 |
| To gain support from my family and others | −0.430 | 0.021 | −0.066 | −1.997 | 0.000 |
| Model summary: $R^2$ = 0.004, F (1, 898) = 3.988, *p* < 0.0005 | | | | | |
| 1 (constant) | 2.433 | 0.050 | 49.097 | | 0.000 |
| To avoid impregnating women or getting pregnant | 0.028 | 0.020 | 0.045 | 1.364 | 0.173 |
| Model summary: $R^2$ = 0.002, F (1, 898) = 1.862, *p* > 0.05 | | | | | |

## 4. Discussion

This study examined the rate of HIV status disclosure and sexual behavior changes and identified factors motivating such changes in sexual behavior among PLHIV on ART in the Northern Region. The findings revealed a high level of HIV-positive status disclosure (60%) among PLHIV on ART in the Northern Region, with no gender difference. Moreover, patients who did not disclose their HIV-positive status did so for a variety of reasons, including fear of others learning of their HIV status, stigma, fear of loss of family support, and non-readiness to do so at the time of the study. More than half (50%) of PLHIV changed their sexual risk behavior "to avoid spreading the disease", "avoid contracting other STIs", "to live a long live", "to conceal their HIV-positive status", "to achieve good ART results", "to live a Godly life", and "to gain support from family" among others.

The increased use of condoms by these HIV-positive patients may be due to the awareness of their positive status and knowledge gained at the ART centers of the preventive methods, such as consistent use of condoms to reduce HIV transmission to their sexual partners. Previous studies have revealed significantly high and consistent use of condoms among PLHIV on ART because of a rise in awareness of the preventive measures, which is more common in male patients, especially when condom use is seen as a male act [16].

Similarly, condom use was revealed as an important strategy comprehensively and sustainably for the prevention of HIV and other STIs, especially among people living with HIV [30]. Therefore, enrolling in ART could be an effective means to encourage positive sexual behavioral change that helps in achieving the agenda 90, 90, 90 by the year 2030 [31]. Thus, the lower rate of condom use is afront and poses difficulty in achieving this agenda nationally and in the Northern Region.

Unfortunately, close to half (45%) of these HIV patients either never or rarely used condoms during sexual intercourse. This outcome may be because of the perceived low sexual gratification and satisfaction associated with the barrier methods of contraceptive use,

such as condom usage. Several individuals may engage in irregular use of condoms during sexual intercourse. A similar finding indicated that about 44% of the participants used condoms irregularly [32]. However, scholarly evidence has already shown that PLHIVs between 60% and 80% are less likely to use a condom during sexual intercourse [33,34]. These disparities could be attributed to the fact that the previous study [33] used fewer than 300 patients who were either on ART or not, whereas the other study [34] focused primarily on females. Thus, results and findings based on 300 participants may be too homogenous to be compared with those of 900. Additionally, since condom use is a predominantly male role, only female participants are likely to reveal such a low level of condom utilization. Moreover, in a male culturally pervasive society, such as in Northern parts of Ghana, it is difficult for females to negotiate condom use [33]. Thus, male clients need more condom use education to protect everyone.

More than half of the patients disclosed their HIV-positive status to their spouses. Perhaps, these patients needed to obtain their partners' support in the ART treatment process, as well as provide their spouses with the opportunity to also test and learn their HIV status. This finding is consistent with that from Nigeria, where 66.3% of HIV-positive clients disclosed their positive status to their spousal partners [35]. It is further shown that HIV-positive status disclosure, especially to spouses, contributes to HIV and AIDS treatment and prevention [24]. Additionally, other findings observed that HIV-positive status disclosure is for emotional and financial support, prevention, family closeness, and the freedom to openly access ART [13,36]. Thus, many of the patients could have disclosed to their spouses for support towards ART [37]. Again, those who reported to their parents and siblings could be doing so in order to obtain their support for the treatment. In support of the above, evidence [38] suggests that disclosure could allow partners to reduce sexual risk behaviors and also make informed future reproductive health decisions, including HIV testing in partners whose status is unknown.

Again, more than half of the participants disclosed their HIV-positive status to their spouses. The reason for this could be that participants needed to obtain their partners' support in the ART treatment process, as well as provide their spouses with the opportunity to also test and learn their HIV status. However, the 40% who did not declare their status may result in a lower percentage of ART treatment-seeking, contributing to a rise in HIV infection rates in the region.

No difference in HIV-positive status disclosure between male and female clients on ART was identified in the Northern Region. This lack of gender disparity in HIV-positive status disclosure is because both male and female patients are almost equally open to discussing their HIV-positive status with others via ART support services [37]. Similarly, studies from southwest Ethiopia among PLHIV showed no difference in the proportion of HIV-positive status disclosure among men and women [39,40]. Contrary to the current finding, a finding from Malaysia [41] revealed that among males, the percentage of disclosure was 66.7% compared with females (70.2%), and the difference was statistically significant. These authors revealed further that men tend to avoid communication, such as HIV disclosure, and that males perceived that disclosure is highly personal information compared with females. They further suggested that HIV-positive men may want to protect their families from the stigma and shame associated with the infection by not revealing their positive status. However, the proportion of males who disclosed being HIV-positive was higher (85.6%) than females (79.5%), though not statistically significant [12]. These disparities could be explained by differences in disclosure targets as well as the study design used in assessing disclosure. That is, the current study used the Chi-square test of independence to establish the gender difference in disclosure rates, whereas the other study [12] used frequencies of self-reported disclosure proportions to reach a conclusion.

Other findings suggest that in the Northern Region, fear of stigma, fear of family neglect, and lack of readiness to disclose HIV-positive status are still high and are influencing patients' decisions to either report their HIV-positive status to close relations. That is, as these factors continue to linger and affect this category of patients in the region, many

patients will continue to conceal their HIV-positive status. Hence, ART treatment-seeking is likely to be low, which may contribute to an increase in the infection rate of HIV in the region. Thus, HIV and AIDS burden on patients, families, the healthcare system, healthcare workers, and other agencies will continue to heighten, and achieving the goal of 90-90-90 would be far-fetched.

## 5. Limitations

This study provides important information regarding positive status disclosure and sexual risk behavior changes among PLHIV in the Northern Region of Ghana. However, it has some limitations. This study is restricted to a single region in Ghana and participants were selected by accidental sampling, which limit the generalisability of the findings to other PLHIV populations beyond the region. The data were collected during the height of the COVID-19 pandemic when many patients were not regularly at ART centers for their drugs due to restrictions on movement or lockdowns placed on the general public. This constraint actually resulted in a low number of clients who visited these healthcare facilities being selected to be part of the sample for this study. Moreover, the disclosure status is relatively high because this is a facility-based study where many of the patients on ART would be encouraged to perform such disclosures to limit reinfection and infecting others. Additionally, because of the sensitive nature of the study, it is likely that studied patients may have given responses suitable to the researchers (social desirability bias). Other patients may also have forgotten vital information because related-issues may have occurred over an extended period (recall bias).

There is a need to explore the reasons some patients do not disclose their HIV status and how healthcare providers can better support patients. Additionally, further research is required on interventions to increase condom use among PLHIV on ART. Further, using a more comprehensive sampling method to ensure the generalizability of findings to a larger population is warranted. A longitudinal study to assess changes in sexual behavior and disclosure rates over an extended period would be ideal. Future studies could benefit from incorporating qualitative methods to better understand the lived-experiences of PLHIV on ART and socio-cultural factors that influence their sexual behaviors and disclosure decisions.

## 6. Conclusions

There was generally a high self-disclosure percentage of HIV-positive status, with more of the patients disclosing either to their spouses or parents. Again, among patients who reported not disclosing their positive status, there was a general fear of stigma and the spread of information about their positive status to others. Furthermore, despite a majority of the clients adopted safe sexual behaviors after learning of their HIV-positive status, a significant number of them were still engaging in high-risk sexual behaviors that could lead to reinfection or the spread of the HIV virus to others. This observation means that to achieve agenda 90, 90, 90 by 2030, and reduce new HIV infections, there is a need to enhance education on safe sexual practices among PLHIV during ART visits. These goals could be achieved in part when public education against stigmatizing PLHIV is intensified in the region. Health workers at ART centers need to constantly remind their clients of the importance of disclosing their status and encourage them to do so to at least one trusted person. It is believed that this approach would go a long way to improve the current 40% non-disclosure rate with its accompanying benefits. Probably, there is a need for a policy to encourage disclosure and the use of "treatment assists" by clients on ART. This suggestion is likely to increase the disclosure of infection status and encourage assistance given by significant others in the course of treatment of PLHIV. Males and females living with HIV need to be offered similar opportunities and support by health workers within the Northern Region to encourage HIV-positive status disclosure. The Ghana Health Service (GHS), through its agencies and workers within the region, needs to intensify public campaigns

against all barriers militating against positive status disclosure among PLHIV on ART within the region.

**Author Contributions:** P.C.K. and E.W.A. conceived the study. P.C.K., E.W.A. and J.E.H.J. designed the study and wrote the literature review and the method. P.C.K. collected the data. P.C.K., E.W.A. and J.E.H.J. performed the data analysis, and P.C.K. wrote the initial draft. All authors have read and agreed to the published version of the manuscript.

**Funding:** This research received no external funding. However, the authors sincerely thank Bielefeld University, Germany for providing financial support through the Institutional Open Access Publication Fund for the Article Processing Charge (APC).

**Institutional Review Board Statement:** This study was approved by the Institutional Review Board (IRB) at the University of Cape Coast, Ghana; ID: UCCIRB/CES/2020/23. We also sought and gained permission from the Northern Regional Director of Health Services and the Heads of all the ART centers. Participants either gave written or verbal consent before taking part in the study. This study was carried out in accordance with relevant guidelines and regulations such as the Helsinki Ethical Principles for Medical Research involving human subject research. We also attached an informed consent form to the instrument, which each participant signed before taking part in the study. Those who could not sign gave verbal consent.

**Informed Consent Statement:** Participants were informed of the possibility that we could publish from the data, but with a high level of confidentiality and anonymity of individually identifiable information.

**Data Availability Statement:** All relevant data are within the manuscript. The datasets used and/or during the current study are available from the corresponding author upon reasonable request. The data are deposited at Open Science Framework and can be found here: https://osf.io/b49ta/ (accessed on 30 April 2023).

**Acknowledgments:** We are grateful to all the participants who spent their valuable time providing us with data for this study. We are equally thankful to the Director of Health Services, Northern Region, and the Heads of the ART clinics for granting us permission to carry out the study at their facilities.

**Conflicts of Interest:** All authors declare that they have no competing interests.

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
