# Peer review of "Positive Status Disclosure and Sexual Risk Behavior Changes among People Living with HIV in the Northern Region of Ghana"

_2036-7449, doi:10.3390/idr15030026_

Round 1

Reviewer 1 Report

Dear Authors,

I had the opportunity to review your manuscript and I appreciate the effort you have put into the study. Your research provides valuable insight into the rate of HIV status disclosure and sexual behavior changes among PLHIV in the Northern Region of Ghana. However, I do have some critical feedback on the manuscript that I would like to share with you.

Introduction

The introduction should be more concise and focused, with a clearer emphasis on the specific research objective and questions.

The introduction should provide more specific information on the prevalence of HIV and ART usage in the Northern Region of Ghana, as well as any regional or population-based variations in HIV prevalence.

The introduction should provide more information on the specific sub-groups of PLHIV on ART in the Northern Region of Ghana that the study will focus on, including any potential sources of bias or confounding.

The introduction could benefit from more explicit connections between the literature review and the study objectives, highlighting how the study will build on previous research and contribute to knowledge in the field.

Methods:

The sampling strategy could have benefited from more information on how the accidental sampling was conducted and the potential sources of bias or confounding that may affect the generalizability of the study findings.

The method section does not provide information on the response rate and any efforts taken to improve response rates.

The questionnaire used in the study could have benefited from more information on the specific items included in each section and how they were designed to address the research questions.

Results:

Overall well written few questions

Clarify the interpretation of some of the regression results, particularly the independent predictors of sexual behavior change. For example, it is not clear how to interpret the finding that "to avoid impregnating women or getting pregnant" was not a significant predictor. Did this mean that it had no effect on behavior change, or that the effect was not statistically significant?

Consider providing more context or explanation for some of the findings. For example, it is not clear why a relatively high percentage of participants (45%) never or rarely used condoms during sexual intercourse, despite having knowledge of their positive status. Providing more insight into the reasons for this behavior could help inform future interventions or counseling strategies.

Discussion:

The discussion section provides a comprehensive overview of the study's findings and their implications. However, some areas of improvement include:

 While the discussion section provides a good overview of the study's findings, it could benefit from a more in-depth analysis of their implications. For instance, the section could discuss the potential impact of the study's findings on the Northern Region's HIV prevention and treatment efforts and how they align with the global agenda of 90-90-90.

The discussion section could also provide suggestions for future research. For example, the section could discuss the need for additional studies to explore the reasons why some patients do not disclose their HIV status and how healthcare providers can better support patients in this regard. Additionally, the section could discuss the need for further research on interventions to increase condom use among PLHIVs on ART.

The discussion section could benefit from a more thorough integration of previous research. For instance, the section could discuss how the study's findings compare to previous studies on HIV status disclosure and sexual behavior changes among PLHIVs on ART in other regions or countries.

By addressing these areas, the discussion section could provide a more nuanced and comprehensive analysis of the study's findings and their implications.

Limitations:

The limitations section of the paper provides a good summary of the potential weaknesses of the study. Accidental sampling may limit the generalizability of the findings to other populations, and the low number of clients visiting health facilities during the COVID-19 pandemic may have limited the sample size and the representativeness of the sample. Additionally, as noted by the authors, the high disclosure rate observed in this study may be partially explained by the fact that the study was conducted in a health facility where patients may have been encouraged to disclose their HIV-positive status. It is important to acknowledge these limitations to ensure that readers interpret the findings within the appropriate context.

One possible suggestion for future research is to use a more representative sampling method to ensure that the findings can be generalized to a larger population. Another recommendation is to conduct a longitudinal study that follows participants over time to assess changes in sexual behavior and disclosure rates over an extended period. This would help to provide more accurate information on how sexual behavior and disclosure change over time, as well as the factors that contribute to these changes. Finally, future studies could also benefit from incorporating qualitative methods to better understand the experiences of PLHIV on ART and the social and cultural factors that influence their sexual behavior and disclosure decisions.

Some other potential limitations that need to be discussed

Social desirability bias as participants may have provided socially desirable responses instead of their actual behaviors and attitudes, particularly when discussing sensitive topics like HIV status disclosure and sexual behavior.

Also recall bias as the study relies on self-reported data, which may be subject to recall bias. Participants may not accurately recall their past behaviors, particularly when it comes to sexual behavior.

Limited generalizability as the study was conducted in a single region of a single country, which may limit its generalizability to other populations or settings.

 The study does not include a control group, making it difficult to determine whether the observed changes in sexual behavior are actually due to ART treatment or other factors.

 The study does not account for other potential confounding variables that may influence HIV status disclosure and sexual behavior changes, such as age, socioeconomic status, and education level.

Conclusion:

The conclusion section could have been more specific in outlining how health workers can enhance education on safe sexual practices during ART visits, as well as how public education against stigmatizing PLHIV can be intensified in the Northern Region. Additionally, the conclusion could have further discussed the implications of the findings for policy and practice, including how the Ghana Health Service can leverage the study findings to design and implement more effective interventions to improve positive status disclosure and sexual risk behavior changes among PLHIV on ART. Overall, the conclusion section provides useful recommendations for future research and interventions, but it could have been strengthened by providing more specific details on how these recommendations can be implemented.

Final comments:

Based on the strengths and weaknesses identified in the manuscript, I would recommend that the manuscript undergo minor revisions before acceptance. The study provides important information on HIV status disclosure and sexual risk behavior among PLHIV on ART in the Northern Region of Ghana. However, there are some limitations and potential areas for improvement that could be addressed through revisions.

Author Response

Please see the attachment for REVIEWER 1

Reviewer 2 Report

This cross sectional study looked into sexual behavior change in people living with HIV on ART. It also looked into voluntary/self  disclosure regarding disease by cases.  The aim is to prevent disease transmission.  Authors have selected group from Ghana which is reasonable given African continent has high prevalence and increased transmission of HIV. Since unprotected sex is one of the common cause of disease

 transmission, authors considered sexual risk behaviour in questionnaire which is valuable. The tool looked into demographic characteristic of patients, percentage of self disclosure, reasons for non disclosure and change in sexual behaviour. These parameters are enough for detailed presentation of identifying barriers for HIV prevention. However only gender difference was assessed in disclosing the HIV status. What about other demographics: employed vs unemployed, single vs married, urban dwellers vs rural, education status. The study mentions looking into the percentage of these demographics, but their effect on disclosure was not reported. Although, predictors for non disclosure attained statistical significance but the odds ratio is less than 1 and confidence interval is wider which speaks of uncertainty in the finding and lack of precision in reported data. May be including larger sample will eliminate this observation.

I concur with the limitations noted in the study that it included group on anti retroviral therapy. But, untreated people living with HIV pose more risk for disease transmission. Overall, the study is clinically relevant and can be instrumental in guidance for curbing HIV pandemic.

Author Response

Please see the attachment for REVIEWER 2

Reviewer 3 Report

This study by Claver et al examines the disclosure status of 900 HIV patients on ART and their reasons for non-disclosure. Because most patients on Art go through counseling and awareness during treatment, their sexual behavioral changes were recorded and statistically analyzed after 3 months of awareness program. As understanding the risk behind such sexual behaviors and subsequently changing it gradually lessens the spread of HIV, this is an essential pattern to monitor. As a reviewer, I believe this was an interesting article and I only have minor comments.

Line 10: Kindly expand “PLHIV” here.

Line 19-21 and line 231-234: Quotation marks are used inconsistently here. The quotation marks can be opened at the start of each predictor and closed at the end of each.

Line 25: “Ghana” may be a better keyword than “northern region”.

Line 36-38: repetitive information. Can be removed.

Line 39: “majority…injection”. This information contradictory to what is mentioned in line 34-35 where the author said that infection due to needle sharing is the least common.

Line 94: Please expand “PLWHA”.

Line 101: “this number” can be replaced by “these” here.

Line 143: I think the author meant “either”.

Line 161: “no formal education” – could those participants read the survey questions? Please mention in methods if any translator were used.

Line 161-162: Is there any specific reason only “Muslim” related demographics is pointed out here (eg. majority of participants were Muslims)? If yes, please provide a justification.

Table 1: Please mention in results section or table caption why is n=538 (and not 540) for some?

Table 3: Was ‘other’ included in the survey? If yes, it should be included in the table as well to present unbiased information. What does ‘constant’ mean here? Also, briefly explain the headings in the caption.

Table 3 and table 4: For the table headings (1st column), the whole words can be written to avoid unnecessary abbreviations.

Line 204-205: “never…had” – was this question only posed to people who had multiple sexual partners before being diagnosed with HIV? Also, was there any selection of participants based on HIV transmission? Were the participants who were asked these questions those who only sexually encountered HIV?

Table 4: I feel that the data can be better presented and will be more valuable to the readers if these variables analysis is done comparing the self-disclosure vs non-disclosure participants.

Line 241: “identify factors motivating such changes in sexual behavior” – the study and analysis does not provide a strong correlation of these yet.

Line 243: The word “difference” here can be replaced with “bias”.

Line 244-246: “patients…reasons” – it is understood from the results that predictors of non-disclosure analysis was done with all participants (n=900) and not just non-disclosure participants. If that is true, this correlation between non-disclosure patients and the reasons for not disclosing can not be made.

Line 250-253: I understand that this sentence is a speculation, but the authors could have confirmed this very easily by adding another survey question asking them about their source of awareness for the increased use of condoms (or any such change in sexual behavior). Is there any reason why this question was not added in the survey?

Line 291-294: This sentence is repetitive and can be removed.

Line 336-337: This first sentence should not be a concluding statement if this is a part of the limitations section. Or it must be properly reworded to explain that this is true only for this specific study.

Author Response

Please see the attachment for REVIEWER 3
